# Visualizing designer quantum states in stable macrocycle quantum corrals

Xinnan Peng [1], Harshitra Mahalingam [2], Shaoqiang Dong[1], Pingo Mutombo [3], Jie Su[1], Mykola Telychko[1], Shaotang Song [1], Pin Lyu[1], Pei Wen Ng[1], Jishan Wu [1], Pavel Jelínek [3,4 ✉], Chunyan Chi[1 ✉], Aleksandr Rodin[2,5 ✉] & Jiong Lu [1,5 ✉]

Creating atomically precise quantum architectures with high digital fidelity and desired quantum states is an important goal in a new era of quantum technology. The strategy of creating these quantum nanostructures mainly relies on atom-by-atom, molecule-by-molecule manipulation or molecular assembly through non-covalent interactions, which thus lack sufficient chemical robustness required for on-chip quantum device operation at elevated temperature. Here, we report a bottom-up synthesis of covalently linked organic quantum corrals (OQCs) with atomic precision to induce the formation of topology-controlled quantum resonance states, arising from a collective interference of scattered electron waves inside the quantum nanocavities. Individual OQCs host a series of atomic orbital-like resonance states whose orbital hybridization into artificial homo-diatomic and hetero-diatomic molecular-like resonance states can be constructed in Cassini oval-shaped OQCs with desired topologies corroborated by joint ab initio and analytic calculations. Our studies open up a new avenue to fabricate covalently linked large-sized OQCs with atomic precision to engineer desired quantum states with high chemical robustness and digital fidelity for future practical applications.

[1] Department of Chemistry, National University of Singapore, Singapore 117543, Singapore. [2] Yale-NUS College, 16 College Avenue West, Singapore 138527, Singapore. [3] Institute of Physics, Czech Academy of Sciences, Prague 16200, Czech Republic. [4] Regional Centre of Advanced Technologies and Materials, Palacký University, Olomouc 78371, Czech Republic. [5] Centre for Advanced 2D Materials (CA2DM), National University of Singapore, Singapore 117543, Singapore. ✉email: jelinekp@fzu.cz; chunyan@nus.edu.sg; aleksandr.rodin@yale-nus.edu.sg; chmluj@nus.edu.sg

Precise engineering of electron wave functions in quantum architectures has offered unprecedented opportunities to not only investigate fundamental aspects of quantum science but also to advance the technological development. The ability to create atomically precise artificial quantum corrals with desired geometries has provided a powerful tool to explore exotic quantum phenomena such as quantum confinement[1], quantum mirage[2,3] and quantum holograph effect[4]. Common strategies for creating artificial quantum nanostructures have mainly involved atomic manipulations and molecular assembly via non-covalent intermolecular interactions. For example, the invention of STM has offered remarkable opportunities to construct new quantum nanostructures out of just a few atoms[5–7] or molecules[8–11] on metal surfaces and directly visualize the quantum states trapped inside these nanocavities. Although tip manipulation enables the fabrication with single-atom precision, these quantum architectures assembled from adatoms or admolecules lack sufficient scalability. One can overcome this scalability limitation by exploiting the supramolecular self-assembly protocols that can lead to the formation of large-sized organic quantum structures assembled via weak non-covalent intermolecular interactions[12–21]. Unfortunately, artificial quantum nanostructures created by these two methods do not have the sufficient chemical robustness required for practical applications.

In contrast, on-surface bottom-up synthesis has revealed its remarkable potential in the fabrication of atomically precise quantum architectures[22,23]. Implementing this approach to fabricate atomically precise covalently linked organic quantum corrals (OQCs) is technologically alluring as it offers high chemical stability, intrinsically digital fidelity, and scalability in materials synthesis. Intensive efforts have been made to synthesize various organic rings with desired geometries and sizeable dimensions[24–27]. However, direct visualization of new quantum states living inside these organic quantum nanocavities remains elusive. This difficulty is partially due to the significant challenge of creating sizeable well-defined OQCs with dimensions comparable to the typical Fermi wavelength of surface electrons on a metallic substrate (e.g. ~3 nm[3,28,29]) to induce the "hot spot" of resonance states arising from the electron scattering due to the potential barrier produced by organic corrals. The synthesis of large-sized organic macrocycles requires delicate control over both thermodynamic and kinetic factors since competitive reaction pathways often yield different side products, and entropy effects disfavour the formation of the ordered rings[30–32].

To this end, we have devised an on-surface synthetic protocol to construct atomically precise covalently linked OQCs from a well-designed organic precursor on Au(111), with the formation of a series of new quantum resonance states, arising from a collective interference of scattered electron waves inside the OQCs. By means of scanning tunneling microscopy, we have directly visualized multiple artificial atomic orbital-like resonance states hosted in individual OQCs, whose orbital hybridization into artificial homo-diatomic and hetero-diatomic molecular-like resonance states can be constructed in OQCs with desired topologies corroborated by joint ab initio and analytic calculations.

## Results

**On-surface synthesis of OQCs.** The bottom-up synthetic route for the fabrication of atomically precise large-sized macrocycle as a quantum resonator involves a chemical design of precursor **1**: 4,4′-((2,6-dimethylphenyl)methylene)bis(bromobenzene). Precursor **1** is expected to undergo the thermally triggered dehydrogenation to create a radical at central methylene position, which can be delocalized to dimethylphenyl for the subsequent

bonding with bromophenyl ring to form a pentagonal ring in **1′** via demethylation[33–35] (refer to the proposed mechanism in Supplementary Fig. 1). As shown in Fig. 1a, two C–Br bonds of precursor **1** with an angle of 120° are expected to facilitate the formation of a 6-unit ring based on pure geometric analysis[36]. The formation of a pentagonal ring in **1′** will enlarge the angle between these two coupling sites to around 150° (equal to the internal angle of dodecagon), thus favouring the formation of large-sized macrocycles with reduced steric-hindrance. Precursor **1** was synthesized via multiple synthetic steps (see "Methods") and then deposited onto Au(111) held at room temperature under ultrahigh vacuum conditions. Subsequently, annealing of Au(111) substrate at 530 K for 20 min triggers the surface-assisted Ullmann coupling and dehydrogenative cyclization (Supplementary Fig. 2) towards the formation of linear polymeric chains, curved ring segments, and perfect 12-unit circular polymer rings (Fig. 1b). Statistical distribution of these curved ring segments and complete 12-unit rings via analysis of multiple STM images (Supplementary Fig. 3) reveal that the yield of curved ring segments tends to be lower with an increase of the number of building units but exhibits one local maximum for the 12-unit rings (2.5%). DFT calculations reveal that the formation energy divided by the number ($n$) of building units decreases for $6 \leq n \leq 12$, while increases for $n \geq 12$. It is noted that a minimum value occurs at $n = 12$ in the plot of formation energies divided by $n$ (Fig. 1c), suggesting that the generation of symmetric 12-unit rings is energetically favourable. A delicate control over the interplay between thermodynamic and kinetic factors may allow for achieving a higher yield of the symmetric 12-unit ring[30–32].

An STM image of the symmetric 12-unit ring acquired by a metallic tip resolves a sunflower-like topology with a pore of 3.86 nm in diameter (Fig. 1b, c). In addition, we also performed bond-resolved STM (BR-STM) imaging using a carbon monoxide functionalized tip (CO-tip) to resolve the internal molecular backbone structure of this macrocycle[22,37–40]. The BR-STM imaging was conducted in constant height mode in the Pauli repulsion regime, wherein the CO molecule undergoes a lateral relaxation over the areas with a high electron density (chemical bond), which modulates the overall conductance in the tunnelling junction, yielding sharp features associated with the chemical bond in the tunnelling current channel[41–43]. The corresponding BR-STM image of this macrocycle reveals interconnected 12 triangular-shaped monomers (Fig. 2a), wherein each building unit contains one pentagonal ring resulting from the demethylation of precursor **1** as discussed above (the DFT-relaxed structure of 12-OQC is shown in Supplementary Fig. 5). The degree of aromaticity of 12-OQC is evaluated by performing nucleus-independent chemical shift calculations (NICS) (Supplementary Fig. 6). The formation of a pentagonal ring in the building unit breaks the structural symmetry and results in the formation of one zigzag side (ZS) and one pentagon-decorated side (PS). It is noted that the majority of monomers tend to connect at the same side (ZS or PS) with neighbouring monomers (Fig. 1a). We also noted that one or a few of monomers in such a 12-unit macrocycle might show the reversed arrangement of ZS and PS sides due to the presence of two possible demethylation sides (Fig. 2a). However, such a structural variation in these isomers shows a negligible variation in the energy positions of frontier orbitals and associated energy gap of 12-unit macrocycles (Fig. 2c and Supplementary Fig. 9a). The large-sized 12-unit macrocycles with a circular potential profile are expected to act as organic quantum corrals (12-OQC) to engineer the designer quantum resonance states. In addition, the covalently bonded nature of 12-OQC exhibits high chemical robustness and stability[44] against thermal- or tip-induced diffusion (often occurring in the adatom-

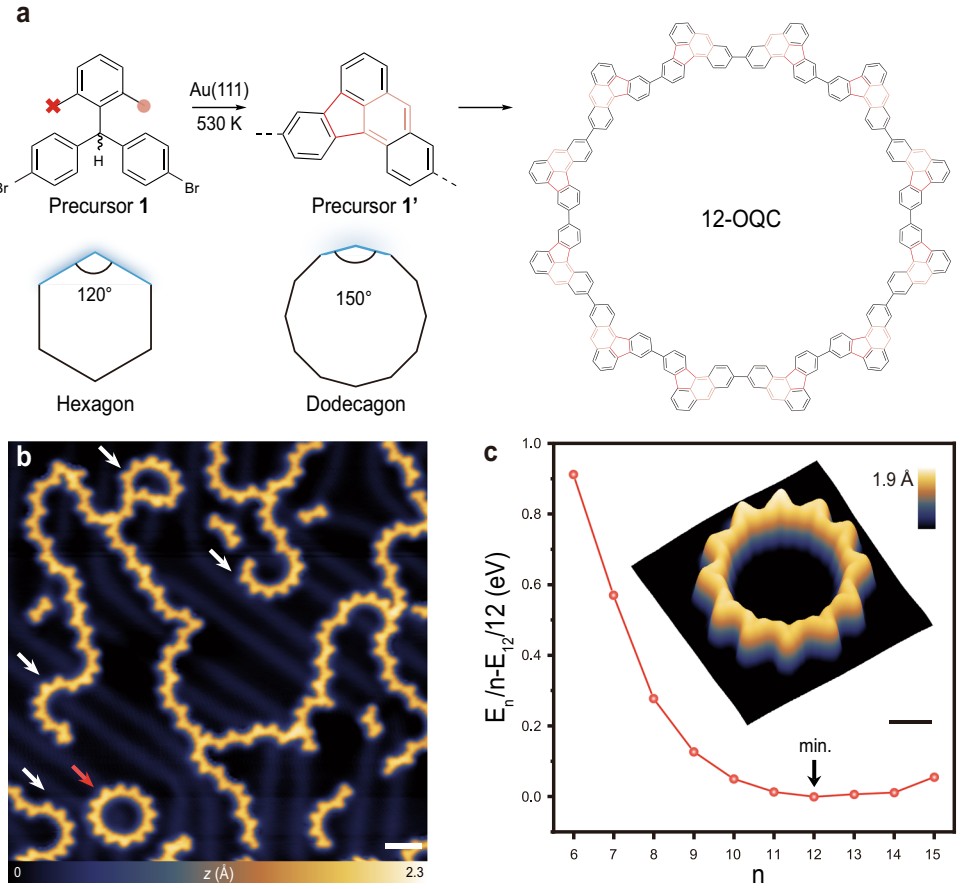

**Fig. 1 On-surface synthesis of covalently linked macrocycles. a** Schematic illustrating the synthetic pathway from precursor (left) to the 12-unit macrocycle (right) with a proposed intermediate (precursor **1'**) with embedded pentagonal ring (middle). The as-formed benzenoid rings (dehydrogenative cyclization) and pentagonal rings (demethylation) are highlighted in orange and red, respectively. Hexagon and dodecagon illustrate the corresponding geometries of polymer rings synthesized from precursor **1** and precursor **1'** (internal angle between the coupling sites is highlighted in blue). **b** Overview STM image after annealing precursor **1** on Au(111) at 530 K, revealing individual 12-unit macrocycle (one of them is highlighted with a red arrow) coexisting with curved segments or half rings (highlighted with white arrows) and polymer chains ($V = 1.0$ V, $I = 200$ pA). **c** Plot of energy differences between total energy divided by number of units and the reference energy of 12-unit macrocycle (zoom-in plot of the region around the energy minimum ($n = 12$) is shown in Supplementary Fig. 4). Inset shows the three-dimensional STM image of one 12-unit macrocycle ($V = -0.1$ V, $I = 200$ pA). The scale bars in (**b**) and (**c**) are 3 and 1 nm, respectively.

derived quantum corrals), making it an ideal candidate to serve as a robust quantum corral.

**Electronic structure of organic corrals.** To probe its electronic structure, we performed differential conductance spectroscopy (d$I$/d$V$) measurement over the 12-OQC on Au(111). Fig. 2c presents the corresponding d$I$/d$V$ spectra collected over the pentagonal ring (blue curve) and the inner edge (red curve) of 12-OQC, along with the reference spectrum recorded on bare Au(111) (grey dashed curve). The d$I$/d$V$ spectrum collected at the pentagonal ring shows two pronounced peaks at $+1.3 \pm 0.05$ and $+1.8 \pm 0.05$ V. In addition, two adjacent peaks at $-1.05 \pm 0.05$ and $-1.45 \pm 0.05$ V appear in the d$I$/d$V$ spectrum acquired at the inner edge of 12-OQC. We further probed the spatial distribution of these molecular states via d$I$/d$V$ mapping with a metal tip. The d$I$/d$V$ maps (Fig. 2d, e) collected at the corresponding energies of $-1.05$ and $+1.3$ V reveal the characteristic nodal patterns predominantly localized at both inner and outer edges, which can be assigned to the valence band (VB) and conduction band (CB), respectively. DFT calculations of a freestanding 12-OQC reveal three nearly degenerate frontier orbitals for both filled and empty states, which yield a band-like electronic structure of this macrocycle (the wave functions of the corresponding frontier orbitals

are shown in Supplementary Fig. 8). The calculated d$I$/d$V$ maps with an s-wave tip (Fig. 2f, g) show good agreement with experimental patterns observed for VB and CB, respectively. Both experimental and theoretical d$I$/d$V$ plots reveal a stronger intensity localized at the PS and ZS for CB and VB, respectively. In addition, we also performed the d$I$/d$V$ measurements over another 12-OQC ring with slightly different internal arrangement of building units as shown in Supplementary Fig. 9, which reveal the energetic positions of CB and VB similar to that of the 12-OQC shown in Fig. 2. Therefore, electronic structures of 12-OQC show a negligible variation even in the presence of misaligned monomers with the opposite ZS/PS arrangement in these macrocyles, which renders them as robust OQCs with high digital fidelity to confine the surface electrons into a series of quantum resonance states as discussed below.

**Quantum resonance states in the 12-OQC.** Interfacial charge redistribution often occurs between molecular adsorbents and metallic surfaces[45,46], resulting in local potential variation over the macrocycle compared to the bare surface. Our DFT calculations also reveal that the charge redistribution leads to the formation of a circular negative potential profile at the organic backbone (Fig. 2b and Supplementary Fig. 7) that can induce

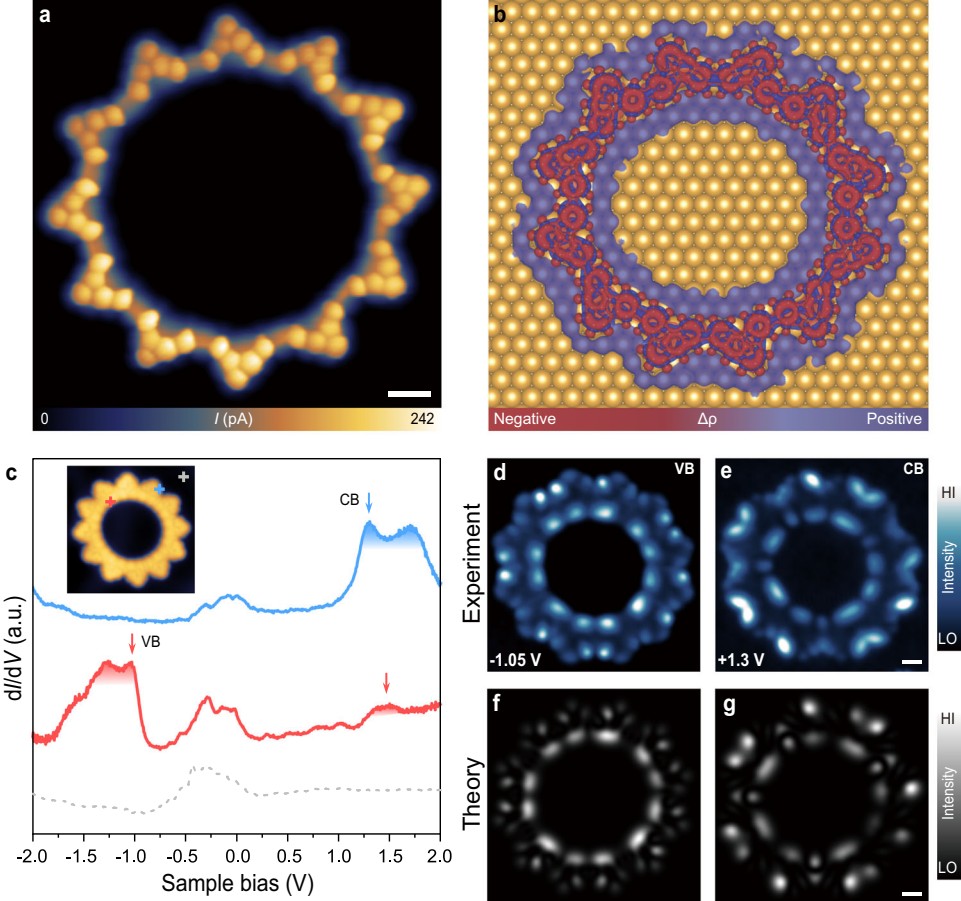

**Fig. 2 Structural and electronic structure characterization of 12-OQC. a** Corresponding BR-STM image ($V = 3$ mV, $\Delta z = -1.2$ Å; set point prior to turn off feedback, $V = 20$ mV, $I = 600$ pA). **b** 3D isosurface of the electronic density differences with an isovalue of $\pm 0.001 e$/Å$^3$ of 12-OQC on Au(111) surface. Red and purple regions in 3D plots represent charge depletion and accumulation, respectively. **c** Point d$I$/d$V$ spectra acquired over different sites of 12-OQC and Au(111) substrate. d$I$/d$V$ curves taken at the pentagonal ring (blue curve), inner edge of 12-OQC (red curve), and taken on Au(111) (grey dashed curve). **d**, **e** Constant-current d$I$/d$V$ maps recorded at the energy positions of the VB ($-1.05$ V) and CB ($+1.3$ V) of 12-OQC, respectively ($I = 2$ nA for **d**, **e**). **f**, **g** Simulated d$I$/d$V$ maps acquired at different energy positions corresponding to different sets of orbitals: **f** $\psi_1$ ($-0.68601$ eV), $\psi_2$ ($-0.67676$ eV), $\psi_3$ ($-0.65554$ eV); **g** $\psi_4$ (1.03913 eV), $\psi_5$ (1.05283 eV), $\psi_6$ (1.06462 eV). Scale bar: 5 Å.

scattering of surface electrons, leading to the formation of quantum states. We then performed d$I$/d$V$ measurements to probe the energy-dependent local density of states (LDOS) inside 12-OQC. As shown in Fig. 3a, the d$I$/d$V$ spectrum (blue curve) acquired at the centre of 12-OQC reveals additional features including a sharp peak at $-0.26 \pm 0.05$ V (labelled as $P_1$), and two broad peaks centred at $+0.46 \pm 0.1$ V (labelled as $P_3$) and $+1.56 \pm 0.1$ V (labelled as $P_5$). Moreover, the d$I$/d$V$ spectrum (red curve) acquired 0.75 nm away from the centre reveals another resonance state at $+0.91 \pm 0.1$ V (labelled as $P_4$). In addition, a weak resonance at $+40 \pm 20$ mV (labelled as $P_2$) along with molecular CB state were revealed in the spectrum (orange curve) acquired near the inner edge of 12-OQC (1.25 nm away from the centre of 12-OQC). We then carried out d$I$/d$V$ mapping to probe the spatial distribution of these new electronic states inside the ring. As shown in Fig. 3d–g, the $P_1$ state exhibits a domelike pattern inside the 12-OQC, in contrast to the dark region right over the 12-OQC at the same sample bias. The $P_3$ state displays a bright dot surrounded by a dark ring, concentrically followed by a bright ring merging with the inner edge of 12-OQC (which can be seen more clearly in simulated spectral function maps in Fig. 3i–k). The d$I$/d$V$ map of the $P_5$ state is characterized by a protrusion in the centre concentrically surrounded by a darker inner and brighter outer ring-like feature. In addition, the dotted

pattern observed over the ring at this sample bias energetically close to that of CB can be assigned to the contribution from the CB state of 12-OQC. Moreover, the d$I$/d$V$ map of the $P_4$ state reveals a doughnut-shaped pattern. Unfortunately, the spatial distribution of the $P_2$ state cannot be resolved clearly by d$I$/d$V$ mapping at constant current mode near Fermi energy due to the presence of molecular topographical variation (Supplementary Fig. 11). The $P_2$ state can be expected to exhibit a doughnut-shaped pattern (similar to that of $P_4$ state) located near the inner edge of 12-OQC as evidenced from the two-dimensional (2D) contour plot of d$I$/d$V$ spectra acquired across the pore of 12-OQC (Fig. 3b).

**Quantum resonance states in the Cassini oval-shaped OQCs.** We also observed the formation of regular Cassini oval-shaped OQC (COS-OQC) (Fig. 4a), which can be viewed as two 6-unit half rings connected by two monomer linkers pointing to the centre, as verified by the corresponding BR-STM image (Supplementary Fig. 12). The COS-OQC with an open channel at the centre allows for the strong electronic coupling between adjacent quantum states which otherwise live in two isolated half-rings. Such a desired topology with electronic coupling is expected to mimic the orbital hybridization effect in homo-diatomic

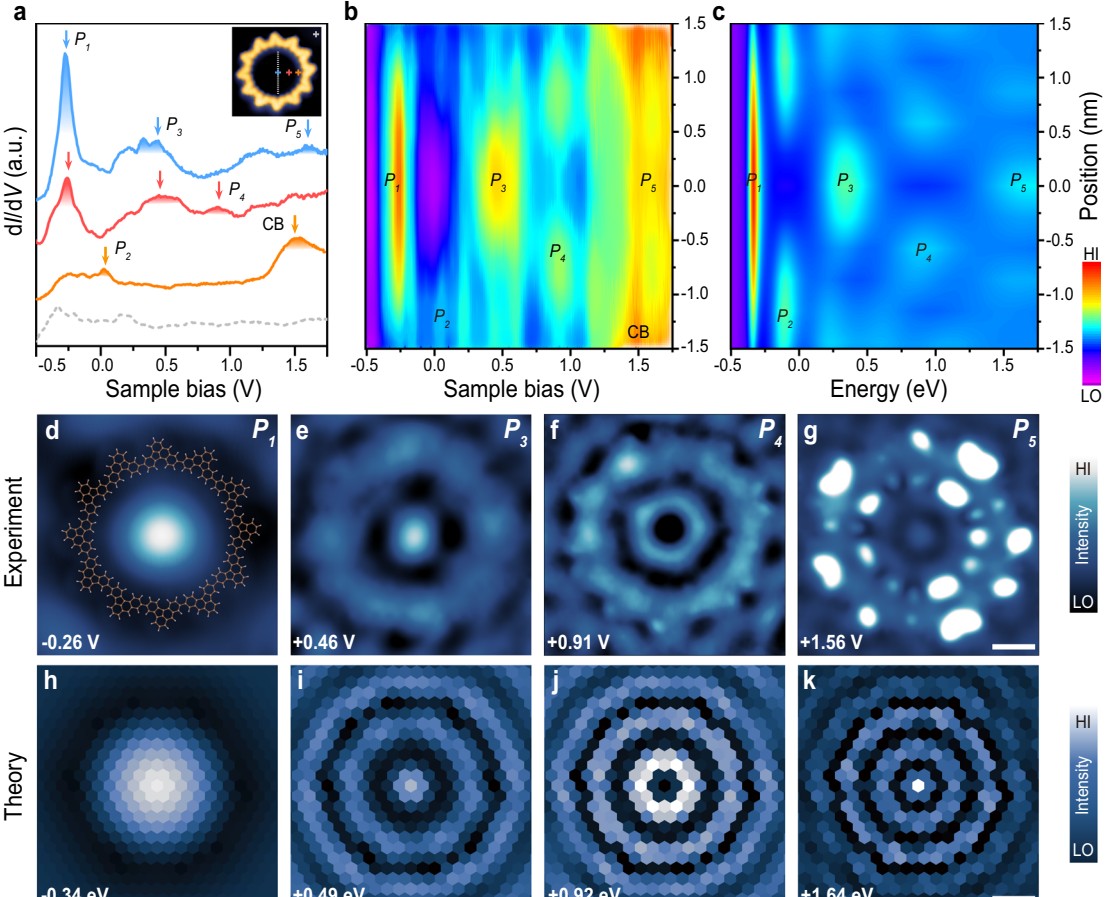

**Fig. 3 Characterization of quantum resonance states trapped in a 12-OQC. a** Point d$I$/d$V$ spectra acquired over different sites inside a 12-OQC and Au(111) substrate. d$I$/d$V$ curves taken at the centre (blue curve), 0.75 nm away from the centre of 12-OQC (red curve), 1.25 nm away from the centre of 12-OQC (orange curve), and taken on Au(111) (grey dashed curve). **b** Colour-coded d$I$/d$V$ spectra (spaced by 0.15 nm) taken across the pore of 12-OQC. The actual positions where the d$I$/d$V$ spectra were taken are indicated by grey dots in the inset STM image in (**a**). **c** Colour-coded simulated spectral function curves (spaced by 0.15 nm) taken across the pore of potential corral in gold unit cells (Supplementary Fig. 16). **d–g** Constant-current d$I$/d$V$ maps recorded at different energy positions ($I = 1$ nA for (**d**, **e**); $I = 1.5$ nA for (**f**, **g**). Panels **d–g** refer to the $P_1$ (−0.26 V), $P_3$ (+0.46 V), $P_4$ (+0.91 V), and $P_5$ (+1.56 V) of resonance states, respectively. (The chemical structure of 12-OQC is superimposed on panel (**d**) to indicate its position.) **h–k** Simulated spectral function maps acquired at different energy positions close to the energetic positions of these quantum resonance states observed experimentally. Panels **h–k** refer to the the $P_1$ (−0.34 eV), $P_3$ (+0.49 eV), $P_4$ (+0.92 eV), and $P_5$ (+1.64 eV) of resonance states, respectively. Scale bar: 1 nm.

molecules. Unlike isolated 12-OQC, a pair of electronic states (Fig. 4b) were observed at −0.32 ± 0.05 and −0.18 ± 0.05 V, predominantly located at the neck of COS-OQC (blue cross) and the centre of two half rings (red cross), respectively. d$I$/d$V$ maps acquired at the energies corresponding to these two emerging electronic states reveal that the low-energy state (−0.32 V) (Fig. 4e) has an oval shape with stronger intensity localized at the neck of COS-OQC. In contrast, the high-energy state (−0.18 V) (Fig. 4f) shows an hourglass-like shape with stronger intensity localized at the centre of half rings with a node pattern at the neck. Such a spatial pattern resembles the characteristic bonding and antibonding states of $H_2$-like molecular orbitals, which is presumably due to the orbital hybridization between adjacent quantum states in two coupled half rings. This will be further discussed in the next section.

## Discussion
**Probing the origin of quantum resonance states in OQCs.** Introducing any perturbation to a 2D electron gas system breaks the translational invariance, giving rise to a position-dependent LDOS. In this study, the gold surface states are perturbed by OQCs. To

investigate the nature of such a perturbation, we first performed ab initio calculations, demonstrating that there is virtually no direct charge transfer between gold and 12-OQC. However, the orbital repulsion between the two components forms an interfacial dipole between the nanoring and the metallic substrate[45,46], resulting in a short-range repulsive potential to which the surface electrons respond (Fig. 2b and Supplementary Fig. 7).

Despite the presence of a circular potential barrier resembling a 2D well, this system does not contain bound states, which can be understood as follows. The potential profile experienced by the surface electrons is limited to the region right below the OQC. The potential inside the OQC is identical to that outside, which can be set to zero. Hence, for an electron inside the corral, the corral boundary represents a potential barrier with finite width and height between two regions of equal potential. This barrier structure guarantees a finite tunnelling probability for the electron, in contrast to a bound state with an infinite lifetime. In fact, solving the Schrödinger equation for a 2D parabolic dispersion with such a barrier does not yield solutions that decay exponentially outside the corral. Instead, one obtains radially propagating Bessel functions, representing free states. Moreover, the model of bound states cannot be used to explain the standing

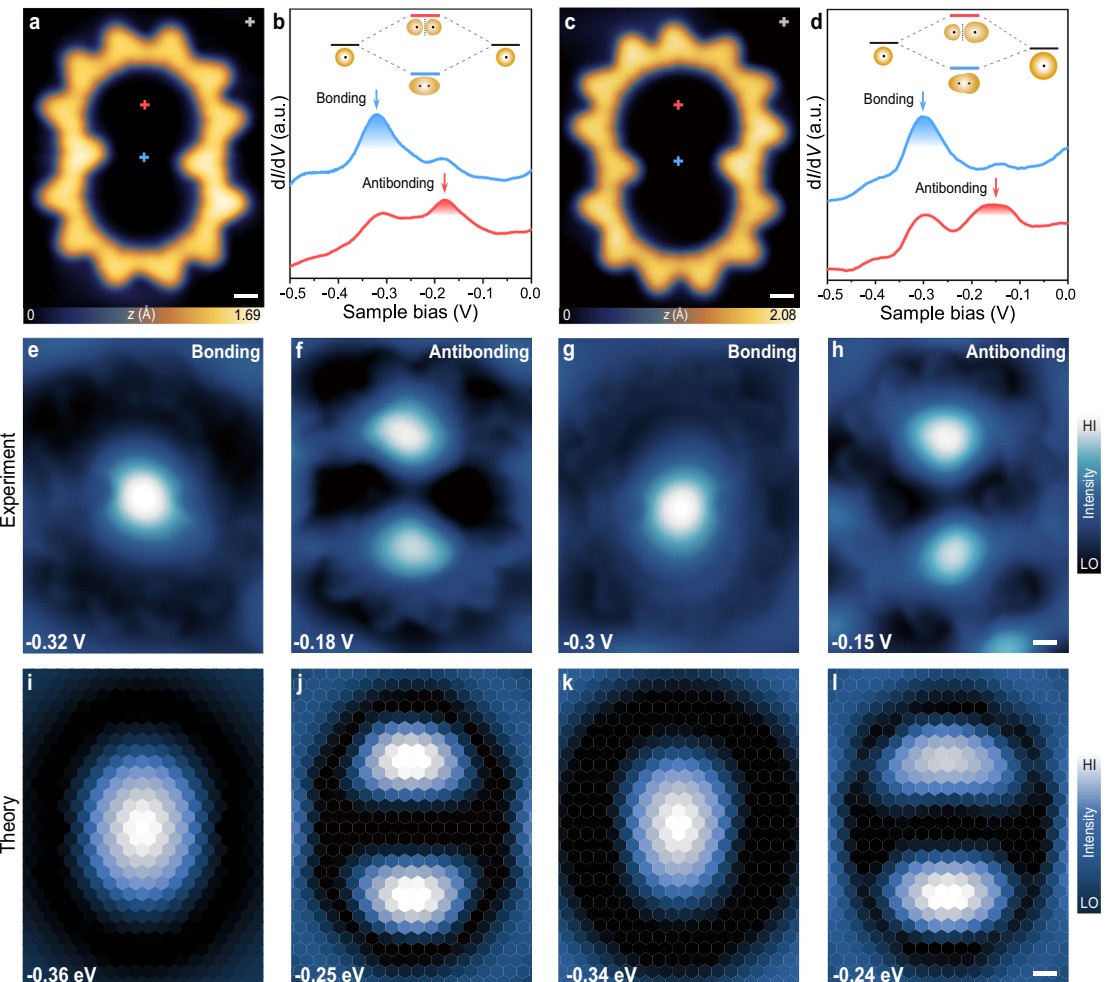

**Fig. 4 Characterization of coupled quantum resonance states in COS-OQCs. a** STM image of a symmetric COS-OQC ($V = -0.32$ V, $I = 1.5$ nA). **b** Point dI/dV spectra acquired over different sites inside a symmetric COS-OQC and Au(111) substrate. dI/dV curves taken at the neck (blue curve) and 1.23 nm away from the neck of symmetric COS-OQC (red curve). **c** STM image of an asymmetric COS-OQC ($V = -0.3$ V, $I = 1.5$ nA). **d** Point dI/dV spectra acquired over different sites inside an asymmetric COS-OQC and Au(111) substrate. dI/dV curves taken at the neck (blue curve) and 1.4 nm away from the neck of an asymmetric COS-OQC (red curve). **e, f** Constant-current dI/dV maps recorded at different energy positions over a symmetric COS-OQC ($I = 1$ nA for **e, f**). Panels **e, f** refer to the bonding ($-0.32$ V) and antibonding ($-0.18$ V) states, respectively. **g, h** Constant-current dI/dV maps recorded at different energy positions over an asymmentric COS-OQC ($I = 1$ nA for **g, h**). Panels **g, h** refer to the bonding ($-0.3$ V) and antibonding ($-0.15$ V) states, respectively. **i, j** Simulated spectral function maps of a symmetric COS-OQC acquired at different energy positions corresponding to bonding ($-0.36$ eV) and antibonding states ($-0.25$ eV), respectively. **k, l** Simulated spectral function maps of an asymmetric COS-OQC acquired at different energy positions corresponding to bonding ($-0.34$ eV) and antibonding states ($-0.24$ eV), respectively. Scale bar: 5 Å.

wave patterns (Supplementary Fig. 14) outside the potential barrier where the surface electrons are totally free.

Although the corral does not produce true confined states, it does give rise to resonances, which can be viewed as hybridizations of true bound states of a circular well with the free-propagating surface electrons[47]. The greater the barrier produced by the corral, the weaker the tunnelling (hybridization) and the more the electronic density inside the corral resembles the confined states (Supplementary Fig. 15). We, therefore, can describe the formation of these new quantum resonances states in 12-OQC and COS-OQC within an electron scattering potential framework, which employs nearly-free electron model to describe the surface states of Au(111). This approach yields a 2D isotropic parabolic dispersion with an effective electronic mass of $0.27m_e$[48,49], where $m_e$ is the mass of an electron, and the Fermi energy is located at 0.46 eV above the band minimum[50,51].

To include the local potential variation, we first partition the gold surface into a grid of unit cells, each hosting a single orbital

of the single-band model. We then include a positive (repulsive) potential term for the unit cells influenced by the dipole from OQCs. Because the dipole potential decays quickly with distance, we only include it for the unit cells directly under OQC (Supplementary Fig. 16). The repulsive potential contour generated by a OQC produces a "corral" for surface electrons.

Introducing a point-like potential perturbation to a metallic system with isotropic dispersion creates the well-known Friedel oscillations in the electronic density with the period determined by the magnitude of the Fermi momentum $|\mathbf{k}_F|$[52–54]. Moreover, electrons at each momentum $|\mathbf{k}| < |\mathbf{k}_F|$ also produce oscillations in the density with longer wavelengths which can be visualized experimentally in dI/dV maps[55]. Including multiple scatterers in the system produces interference, which can be treated as a superposition of the scattering patterns from each point scatter considered individually. Alternatively, one can employ a non-perturbative approach and calculate the resultant density variation for a collection of scatterers simultaneously. In this

study, we make use of the latter approach, as discussed in the Supplementary Note 1.

To compare theoretical results to the experimental data, we first "draw" the corral of the appropriate shape by introducing local repulsive potential to the gold unit cells located below the OQC. Next, we calculate the position-dependent spectral function following the procedure described in Supplementary Note 1. The spectral function gives the density of states at a particular unit cell at a given energy. Because the differential conductance at some bias $V$ is related to the number of states available for tunnelling at that energy, we use the spectral function as a proxy for the d$I$/d$V$ data. By fixing the energy at which the spectral function is computed, it is possible to obtain spectral function spatial maps like the ones shown in Figs. 3h–k and 4i–l. On the other hand, if the position is kept constant, varying the energy yields the d$I$/d$V$-like curves (Supplementary Fig. 10), corresponding to horizontal slices of Fig. 3c. Because of the generality of our formalism, we can study the highly symmetric circular OQC and the more challenging COS-OQC.

The spectral function depends on the electronic dispersion and the scattering potential. Assuming that the potential produced by OQC is restricted to the region directly below the polymer, our theoretical treatment only requires a single fitting parameter: the magnitude of this potential because we already know the effective mass of the surface-state electrons. As was discussed above, the greater the potential barrier separating the interior of the OQC from the rest of the surface, the more the resonances resemble the true bound states. Reducing the potential broadens the states, as shown in Supplementary Fig. 15.

With a repulsive potential of 0.6 eV, the simulated spectral function curves and 2D contour plot of spectral function curves (Fig. 3c) show good agreement with experimental data. Such a value, lower than the typical amount of charge transferred in the molecular systems with the occupation and depletion of molecular states, can therefore be rationalized in our system only with interfacial charge polarization, consistent with our DFT predictions. We also noticed several mild discrepancies between experimental d$I$/d$V$ data (Fig. 3b) and the theoretical simulations (Fig. 3c). First, at the high sample bias (~1.5 V), the experimental results contain the signature of the CB of 12-OQC that is not included in the theoretical model. Additionally, the peaks in the experimental data are shifted towards slightly higher bias compared to the theoretical ones. This mild quantitative disagreement is not unexpected because, as was stated above, the spectral function is a proxy for the d$I$/d$V$ signal with the latter depending also on the electronic convolution with the tip. The key features, such as the d$I$/d$V$ peaks and the oscillating signal rings in the d$I$/d$V$ maps, however, are robust and show good agreement between theory and experiment.

Next, we discuss the hybridization phenomenon in COS-OQCs. The 2D electron gas (2DEG) with discrete resonance states could be viewed as an artificial "hydrogen atom" constructed by the circular corral. In principle, artificial homo-diatomic $H_2$-like molecular orbitals can be realized in COS-OQC, wherein two adjacent "H atoms" can undergo orbital hybridization. Firstly, the energy halfway between the two emerging electronic states in COS-OQC (−0.25 eV) is close to the energy position of $P_1$ state (−0.26 eV) in a 12-OQC because of their comparable diameters. It could be envisioned that the original "atomic" orbitals constructed in the two half rings with open neck channel can effectively hybridize and split into one bonding state at lower energy and one antibonding state at higher energy (inset of Fig. 4b), analogous to the hybridization of two neighbouring resonance states arising from vacancies in molecular self-assembly[56]. The theoretical results, shown in Fig. 4i, j, exhibit a very similar behaviour with the energies of the single-hot spot

and the hourglass-like states sandwiching the $P_1$ peak of the circular OQC.

In addition, we also investigated an asymmetric COS-OQC including a larger half-ring with one more monomer than an adjacent smaller half ring (Fig. 4c). In this case, an artificial "hetero-diatomic molecule" would be constructed due to the energy level mismatch between the original "atomic" orbitals from the two half rings with different sizes. One pair of bonding (−0.3 V)/antibonding (−0.15 V) states and their corresponding d$I$/d$V$ maps are shown in Fig. 4d, g, h, respectively. It is noted that the bonding and antibonding states show asymmetric spatial distribution in the two half rings which agree with the theoretical results in Fig. 4k–l. Therefore, an artificial giant "homo-diatomic molecule" and "hetero-diatomic molecule" can be constructed by engineering the topology of OQCs. Moreover, the tip manipulation can be used to close and open up the OQC to engineer multiple quantum resonance states by controlling their dimensions and geometries (Supplementary Fig. 17). A series of resonance states with hot-spots and ring-like patterns were revealed by d$I$/d$V$ measurements of the irregular rings before and after tip manipulation. As expected, we also observed that these resonance states shift towards higher energy positions as the ring is closed up by tip manipulation, equivalent to the reduction of the lateral dimensions of the OQCs. This opens up virtually unlimited opportunities for engineering the desired quantum resonance states inside chemically robust OQCs.

In summary, we have demonstrated a bottom-up atomically precise synthesis of chemically robust OQCs with topology-controlled new quantum resonance states, arising from a collective interference of scattered electron waves inside the quantum corrals. Individual OQCs with a series of resonance states behave like artificial atomic orbitals. The effective coupling of artificial atomic orbitals leads to the formation of artificial homo-diatomic and hetero-diatomic molecular states in OQCs with desired topologies corroborated by a joint ab initio and analytic calculations. The fabrication of covalently linked large-sized OQCs with atomic precision not only grants access into the quantum nature of these systems with intrinsically digital fidelity but also enables the precise engineering of desired quantum states. The scalable fabrication of artificial quantum nanostructures with high chemical robustness paves the way towards future technological implementation.

## Methods

**Synthesis of molecular precursor.** Precursor **1** was synthesized through the reaction of 2-lithium-1,3-dimethylbenzene with bis(4-bromophenyl)methanone followed by acidification and two-step dehydroxylation reaction. The structure of **1** was further confirmed by [1]H NMR, [13]C NMR and high-resolution GC-Mass. The detailed chemical synthesis procedure of precursor **1** and solution characterization data are shown in Supplementary Figs. 18–21.

**Sample preparation and STM/STS measurements.** The STM experiments were conducted in UHV conditions (base pressure, $<2 \times 10^{-9}$ mbar) at 4.4 K using a Scienta Omicron LT-STM system. Au(111) single crystal (MaTeck GmbH) was cleaned by multiple cycles of Ar$^+$ sputtering ($1 \times 10^{-5}$ mbar) and annealing (710 K, 10 min). The precursor **1** was deposited from Knudsen cell (MBE-Komponenten GmbH) at 360 K onto clean Au(111) surface held at room temperature. After the deposition of the precursor molecules, the sample was annealed at 530 K for 20 min for the fabrication of OQCs. Subsequently, the sample was transferred into the STM/AFM head held at 4.5 K for STM imaging and characterization. All the BR-STM images were collected in constant height mode. The tip-sample distance with respect to an STM set point is indicated in the figure caption for each BR-STM image. The tip apex was functionalized with a CO molecule by picking up CO from Au(111) surface. The d$I$/d$V$ spectra were collected using an internal lock-in amplifier with a modulation frequency of 693 Hz and amplitude of 20 mV. All the d$I$/d$V$ maps were collected in a constant-current mode.

**DFT simulations.** The large-scale total energy DFT calculations were performed using the Fireball package[57]. All geometry optimizations and electronic structure analyses were performed using BLYP exchange-correlation functional[58] with D3

corrections[59] and norm-conserving pseudopotentials with a basis set of optimized numerical atomic-like orbitals[60]. Atomic configuration was allowed to relax until the remaining atomic forces reached below $5 \times 10^{-2}$ eV Å$^{-1}$. We employed slabs model consisting of $60 \times 60$ unit cell of Au(111) surface with two layers and 12-ring molecule, in total 1368 atoms. The last Au layer was kept fixed in the bulk position. Brillouin reciprocal zone was sampled by only $\Gamma$ k-point. The theoretical $dI/dV$ simulations were carried out using probe particle STM code[41,43] using electronic structure of free-standing 12-OQC. Metallic-tip was mimicked by $s$-like orbital without tip relaxation.

**QFT calculations**. The derivation of the field-theoretic formalism is provided in Supplementary Note 1. The numerical calculations were performed using JULIA programming language[61].

## Data availability
All data needed to evaluate the conclusions of this study are available in the main text or Supplementary Information. The data that support the findings of this study are available from the corresponding authors on reasonable request.

## Code availability
The code used in the calculations is available on GitHub at https://github.com/rodin-physics/au-polymer[62].

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

## Acknowledgements

J.L. acknowledges support from MOE grants: MOE2019-T2-2-044 and R-143-000-B58-114. A.R. acknowledges the National Research Foundation, Prime Minister Office, Singapore, under its Medium Sized Centre Programme and the support by Yale-NUS College (through Grant No. R-607-265-380-121). C.C. acknowledges support from the MOE Tier 1 grant (R-143-000-B62-114) and Tier 2 grant (MOE2018-T2-1-152). P.M. and P.J. acknowledge support from the Praemium Academie of the Academy of Science of the Czech Republic and the CzechNanoLab Research Infrastructure supported by MEYS CR (LM2018110). P.J. acknowledges the support of the GACR 20-13692X. P.M. acknowledges MSMT Project No. SOLID21-CZ.02.1.01/0.0/0.0/16_019/0000760.

## Author contributions

X.P., H.M., S.D. and P.M. contributed equally to this work. J.L. supervised the project. X.P., A.R. and J.L. conceived the project. X.P. performed experiments related to on-surface synthesis, STM/STS measurements, and data analysis. J.S., M.T., S.S. and P.L. assisted in the data analysis and contributed to the scientific discussion. S.D., P.W.N., J.W. and C.C. designed and synthesized the precursor molecules; P.M. and P.J. performed the DFT calculations; H.M. performed field theoretic calculations under the supervision of A.R.; The manuscript was written by X.P., H.M., A.R. and J.L. with contributions from all the co-authors.

## Competing interests

The authors declare no competing interests.
