## [Peer Review File · Nature Communications]

REVIEWER COMMENTS

Reviewer #1 (Remarks to the Author):

This paper describes an innovative approach on building quantum corrals from precursor molecular structures on surfaces and the building of molecules.

The visualization of bonding and antibonding states is very impressive and the agreement between experiment and theory is convincing.

The only shortcoming is the missing explanation of the orbitals ψ_{1-3} and ψ_{4-6} in Fig. 2. What exactly does the index of the wave function mean? Is this just a sorting index? It would help to explain these ψ 's a bit more. Can they be approximated by analytical functions to help to imagine their properties?

The same ambiguity occurs in Fig. S3, where $\psi_1 - \psi_3$ could be read as the difference between those wave functions, while the authors probably refer to the DOS of $|\psi_1|^2 + |\psi_2|^2 + |\psi_3|^2$.

Other than that, I think the paper is fine and Nature Communications is the right journal for its publication

Reviewer #2 (Remarks to the Author):

Lu et al report an interesting work of studying confined surface states in covalently linked organic quantum corrals (OQCs). Besides observing quantum states inside a nanocavity provided by a single ring structure, they also observed molecular-like bonding and anti-bonding states in Cassini oval-shaped OQCs. Overall, the data are in high quality and the analysis is also solid. However, the physics of this work is kind of trivial. The claim of "Our studies open up a new avenue to fabricate covalently linked large-sized OQCs with atomic precision to engineer desired quantum states with high chemical robustness and digital fidelity for new-generation quantum technology", is very vague. The reviewer does not know what specific "quantum technology" or "nanophotonics" can emerge from this work since the authors do not elaborate at all.

Reviewer #3 (Remarks to the Author):

The authors present a bottom-up on-surface synthesis of organic quantum corrals (OQC) on Au(111) and its subsequent characterization via LT-STM measurements.

STM images show a series of bonding & antibonding states which are attributed to the hybridization of pseudo-atomic orbitals within the 12-OQC.

Some theoretical and DFT calculations were also performed to better understand the observations.

The study is a nice piece of work, the STM images look great, and the methodology is sound.

It merits publication in Nat. Comm. once the authors address satisfactorily the following issues. Therefore, major revision is recommended here.

Major points:

=====

1. The (tertiary) triphenylmethyl radical (triarylmethyl radical

2 in Scheme 1) is known to be a metastable persistent radical stabilized by pi conjugation/resonance in solution. The high reactivity of the monomer 2 (stated in the caption of Scheme 1) might be actually due to the carbon radicals generated upon C-Br cleavage, see point 5 below.

2. Can the authors comment on the possible aromaticity (or lack thereof) of the macrocycle and its monomer 1'?

A NICS calculation (or another aromaticity local descriptor) on a hydrogenated version of monomer 1' (Fig.1) would be sufficient.

Do you think this is somehow connected to the optimal stability for n=12? or it is just the proposed geometry argument (internal angles of 150° between monomers)?

3. A zoom inset of the region around the minimum (n=12) in Fig.1c is needed. Right now in Fig.1c it is hard to see the energy difference of neighboring points (n=10,11,13,14) relative to the global minimum (n=12). The bottom of the well in Fig.1c looks flat on the vertical scale. If these energy

differences are too small, the used force threshold of 0.05 eV/ÅÅ in the DFT structural relaxations might be on the loose side to converge this properly.

4. I wonder if the authors could discriminate the different bond orders in their BR-STM measurements of the 12-OQC? This is to check agreement with Clar's rule (disjoint aromatic pi sextets) on polycyclic aromatic hydrocarbons (PAH). AFM would have been a more useful technique here though.

5. On-surface debromination of organic adsorbates is known to occur readily on coinage metal surfaces even at room T.

According to the experimental details, precursor 1 was deposited at 360 K onto clean Au(111) surface held at room temperature.

I think on-surface debromination may occur here before or about the same time as the mechanism shown in Scheme 1.

Did authors track the debromination after precursor deposition? On STM, Br adatoms should appear as small protrusions on the surface near the debrominated precursor. A high-resolution STM image (if available) of the isolated (debrominated) precursor 1 on Au(111) could be helpful here to firmly establish the time ordering of the reaction events.

Minor points:

=====

6. The transformation from precursor 1 to precursor 1' (Fig.1a) is some kind of Ullmann coupling reaction. Authors should mention this in the paper.

7. Can authors comment on the adsorption geometry relative to the herringbone reconstruction & the planarity of 12-OQC? I assume that the 12-OQC macrocycle is physisorbed on Au(111). If so, what is the mean vertical height above top layer according to their DFT calculations? From Fig.S2a it seems that 12-OQC lies flat about 3 ÅÅ above top layer. The authors should probably give the DFT-relaxed structures of 12-OQC as a Supplementary Materials for the sake of reproducibility.

8. I wonder whether a simple model of a quantum particle in an annular disk could be helpful to understand the shape of the electronics states in Figs.2 f & g of the freestanding 12-OQC? The solution of the 2D-Schroedinger equation should be analytic yielding Bessel functions of the 1st kind.

9. In Fig.2 f & g, authors should give the calculated energy range of these DFT orbitals. Fig.S3 caption should give the energies of individual Kohn-Sham orbitals and provide the isosurface value of the representation.

10. Did the authors cite all relevant literature about the use of organic molecules in quantum corrals?

On a quick Google search I found (for example) J. Phys. Chem. Lett 7 (16), 3073 (2016) which was not cited here.

11. Perhaps an electron localization functon(ELF) plot of the freestanding 12-OQC would be helpful for point #2.

12. AFM measurements/simulations would have made this study more complete as is usual in this field of surface science. Any comment from the authors about this?

Response to Reviewer for manuscript NCOMMS-21-21515-T

Comments in black - Replies in blue - Amendments to the manuscript in red

Reviewer #1:

This paper describes an innovative approach on building quantum corrals from precursor molecular structures on surfaces and the building of molecules.

The visualization of bonding and antibonding states is very impressive and the agreement between experiment and theory is convincing.

We highly appreciate reviewer for positive comments of our work and recommendation for publication. We have clarified all the open questions below.

The only shortcoming is the missing explanation of the orbitals ψ_{1-3} and ψ_{4-6} in Fig. 2. What exactly does the index of the wave function mean? Is this just a sorting index? It would help to explain these ψ 's a bit more. Can they be approximated by analytical functions to help to imagine their properties?

The same ambiguity occurs in Fig. S3, where $\psi_{1} - \psi_{3}$ could be read as the difference between those wave functions, while the authors probably refer to the DOS of $abs(\psi_{1})^2 + abs(\psi_{2})^2 + abs(\psi_{3})^2$.

Apology for this misunderstanding. We have provided detailed information and explanation to clarify this point. The subscript for wave functions ($\psi_1, \psi_2, \psi_3 \dots$) is just a sorting index. Our DFT calculations of a freestanding 12-OQC reveal three nearly degenerate frontier orbitals for both filled (ψ_1, ψ_2 , and ψ_3) and empty (ψ_4, ψ_5 , and ψ_6) states, which yield a band-like electronic structure of this macrocycle. Simulated dI/dV maps of VB and CB (Fig. 2f,g in the main text) were acquired at different energy positions corresponding to different sets of orbitals (VB from ψ_1, ψ_2 , and ψ_3 ; CB from ψ_4, ψ_5 , and ψ_6). The calculated PDOS and spatial distribution of the six frontier orbitals are shown in Fig. S7 (previous Fig. S3).

We also tested the analytical functions, which show that these electronic states cannot be approximated by the model of a particle in an annular disk due to the following reasons.

(i) Solving the 2D Schrödinger's equation does indeed yield Bessel functions of the first and second kind. We used the asymptotic forms of the Bessel functions as well as the inner and outer radii of the 12-OQC to obtain $E_n \sim Ryd \times \left(\frac{n}{3}\right)^2$ as the approximate form of the energy spacing. From this, we observe that the energy separations between calculated levels are much bigger than the separations between, for example, the VB and CB of 12-OQC.

(ii) In addition, the experimental molecular states show a structure-dependent localization: for example, dI/dV maps (Fig. 2f,g in the main text) reveal a stronger intensity localized at the ZS (zigzag side) and PS (pentagon-decorated side) for CB and VB, respectively. The model of a smooth-walled annular disk fails to capture these features, as seen by the model's lower-lying eigenstates (see below).

Figure R1. Eigenstates of particle in an annular disk for $l = 0, 1$ and $n = 1, 2, 3$.

From our analysis, we feel that the model of a particle in an annular disk is not able to adequately describe the observed electronic states, which is most likely contributed from a collection of nearly degenerate molecular frontier orbitals.

ψ_{1-3} in Fig. S7 (previous Fig. S3) refers to three nearly degenerate filled frontier orbitals: ψ_1, ψ_2, ψ_3 .

ψ_{4-6} in Fig. S7 (previous Fig. S3) refers to three nearly degenerate empty frontier orbitals: ψ_4, ψ_5, ψ_6 .

To avoid the confusion, we have replaced ψ_{1-3} (ψ_{4-6}) with ψ_1, ψ_2, ψ_3 (ψ_4, ψ_5, ψ_6) in both main text and SI.

Actions: We have revised the labels of wave function orbitals in Fig. 2 and Fig. S7 (previous Fig. S3) in the revised manuscript.

Other than that, I think the paper is fine and Nature Communications is the right journal for its publication

Reviewer #2:

Lu et al report an interesting work of studying confined surface states in covalently linked organic quantum corrals (OQCs). Besides observing quantum states inside a nanocavity provided by a single ring structure, they also observed molecular-like bonding and anti-bonding states in Cassini oval-shaped OQCs. Overall, the data are in high quality and the analysis is also solid. However, the physics of this work is kind of trivial. The claim of “Our studies open up a new avenue to fabricate covalently linked large-sized OQCs with atomic precision to engineer desired quantum states with high chemical robustness and digital fidelity for new-generation quantum technology”, is very vague. The reviewer does not know what specific “quantum technology” or “nanophotonics” can emerge from this work since the authors do not elaborate at all.

We thank reviewer for both positive comments and questions raised below for further improvement. We agree with reviewer that the physical picture describing these resonance states inside the OQCs is not new. The novelty of this work lies in fabricating robust large-

sized OQCs with well-defined geometries to explore the designer quantum states and their electronic coupling in novel quantum nanostructures.

Previous work has pointed out that quantum corrals hold a great potential in quantum information technology. As demonstrated by Manoharan et al., the design of elliptical quantum corrals provides an ideal playground to remotely probe information at the atomic scale based on the quantum mirages effect (Nature 403, 512 (2000)). One can not only remotely probe information but also control it using atomically-precise quantum corrals. In addition, the turning on/off probability of nanoscale switches can be increased by remotely applying a pulse inside a quantum corral (Nano Lett. 17, 8 (2017)). Moreover, pseudo basic logic operations, such as NOT, FANOUT and OR gates have been realized by manipulating quantum mirages in quantum corrals (Nature Commun. 11, 1400 (2020)). Future technological implementation of these quantum materials has to meet the following key criteria: high digital fidelity (atomic precision), chemical robustness of artificial quantum architectures, and the scalability in the quantum nanostructure fabrication. Unfortunately, all the current approaches developed to date, such as atomic manipulation in an atom-by-atom or molecule-by-molecule manner or molecular assembly *via* non-covalent intermolecular interactions, fail to meet all these critical requirements. In this work, we would like to highlight the potential scalable fabrication of carbon-based artificial quantum nanostructures *via* on-surface bottom-up synthetic protocol. In addition, atomically-precise covalently-linked OQCs synthesized here are technologically alluring as they offer both high chemical stability and digital fidelity required for quantum technology applications as discussed above.

In view of reviewer's comments, we have revised the relevant sentences in the abstract and conclusion of main text accordingly to tone down the speculative arguments related to their practical applications in quantum technologies.

Actions: We have modified the abstract and conclusion in the revised manuscript.

We have changed the last sentence in abstract into ‘Our studies open up a new avenue to fabricate covalently linked large-sized OQCs with atomic precision to engineer desired quantum states with high chemical robustness and digital fidelity for future practical applications.’

We have changed the last sentence in conclusion into ‘The fabrication of covalently linked large-sized OQCs with atomic precision not only grants access into the quantum nature of these systems with intrinsically digital fidelity but also enables the precise engineering of desired quantum states. The scalable fabrication of artificial quantum nanostructures with high chemical robustness paves the way towards future technological implementation.’

Reviewer #3:

The authors present a bottom-up on-surface synthesis of organic quantum corrals (OQC) on Au(111) and its subsequent characterization via LT-STM measurements.

STM images show a series of bonding & antibonding states which are attributed to the hybridization of pseudo-atomic orbitals within the 12-OQC.

Some theoretical and DFT calculations were also performed to better understand the observations.

The study is a nice piece of work, the STM images look great, and the methodology is sound.

It merits publication in Nat. Comm. once the authors address satisfactorily the following issues. Therefore, major revision is recommended here.

We highly appreciate the reviewer's valuable comments and the support for the publication of our work in Nature Communications after revision. As described below, we have clarified all the open points indicated by the reviewer. Specific actions are described in the point-by-point responses below.

Major points:

=====

1. The (tertiary) triphenylmethyl radical (triarylmethyl radical **2** in Scheme 1) is known to be a metastable persistent radical stabilized by pi conjugation/resonance in solution. The high reactivity of the monomer **2** (stated in the caption of Scheme 1) might be actually due to the carbon radicals generated upon C-Br cleavage, see point 5 below.

We agree with reviewer that radical **2** in Scheme 1 could be metastable persistent radical stabilized by pi conjugation/resonance in solution or even on surface. However, the formation of the pentagonal ring in radical **2** on a catalytically active substrate can still proceed upon annealing. In general, the carbon radicals created upon C-Br cleavage are essential for the polymerization process *via* Ullmann coupling. The high reactivity of the carbon radicals as pointed out by reviewer may be also a possible reason for the demethylation.

The formation of pentagonal rings from demethylation was also observed in a precursor with chemical structure similar to ours (Nature Commun. 11, 6076 (2020)). However, the pentagonal ring is likely to be formed after annealing at a much higher temperature (~673 K), while the demethylation occurs at a relatively low temperature (~450 K) in our case. In addition, the debromination process usually occurs at a lower temperature than that of dehydrogenated cyclization (Nature Commun. 11, 6076 (2020)). Based on these points, we assumed that the delocalized radical from centre carbon could promote the formation of pentagonal rings (nearly 100 % yield), which is essential for the on-surface synthesis of macrocycles reported here.

2. Can the authors comment on the possible aromaticity (or lack thereof) of the macrocycle and its monomer **1'**?

A NICS calculation (or another aromaticity local descriptor) on a hydrogenated version of monomer **1'** (Fig.1) would be sufficient.

Do you think this is somehow connected to the optimal stability for n=12? or it is just the proposed geometry argument (internal angles of 150° between monomers)?

Figure R2. NICS calculations of a hydrogenated version of monomer **1'** (left) and 12-unit macrocycle (right).

This is a good point. To better understand the aromaticity of monomer **1'** and macrocycle, we conducted NICS calculations of the hydrogenated version of **1'** and macrocycle. For the macrocycle, only the repeating unit (two monomers head-to-head linked together) was calculated for simplicity. It shows that all the four phenyl rings of monomer **1'** are aromatic with NICS(1)_{zz} values from -19.90 to -28.40 and the pentagon is antiaromatic with a value of 10.29. The repeating unit of the macrocycle shows slightly decreased aromaticity of four phenyl rings with NICS(1)_{zz} values from -16.80 to -26.87 and remained antiaromatic of pentagons with values of 10.49 and 11.48. In general, the monomer and macrocycle show similar NICS values and localized aromaticity. As expected, the macrocycle does not offer global aromaticity. Aromaticity may play a very limited role in the formation of the 12-unit macrocycles in our case.

The optimal stability for $n=12$ could be attributed to multiple factors, such as geometry and thermodynamics. The macrocycle could be regarded as a stable cyclic oligomer formed by C-C coupling reactions on surface. Therefore, the key issue for the formation of dodecagons is a suitable geometry angle of 150° , which was formed by the 3 and 10 linked positions of monomer **1'**. In addition to the geometry analysis, the formation of 12-unit macrocycles was proven to be energetically favourable based on our DFT calculation of energy differences for macrocycles with different unit numbers (Fig. 1c in the main text and Fig. S3).

Actions: We have placed Fig. R2 in SI (Fig. S5) with the corresponding description.

3. A zoom inset of the region around the minimum ($n=12$) in Fig.1c is needed. Right now in Fig.1c it is hard to see the energy difference of neighboring points ($n=10,11,13,14$) relative to the global minimum ($n=12$). The bottom of the well in Fig.1c looks flat on the vertical scale. If these energy differences are too small, the used force threshold of 0.05 eV/Å in the DFT structural relaxations might be on the loose side to converge this properly.

We agree with reviewer that the energy differences around this region ($10 < n < 14$) are small. To check the convergence criteria, we carried out the total energy DFT simulations with more tight convergence criteria (force: $F_{\text{tol}} = 0.01 \text{ eV/Å}$; total energy threshold: $E_{\text{tol}} = 10^{-5} \text{ eV}$) and

the calculated energies barely change. Therefore, we think that these energy differences are converged with the accuracy of DFT method. The calculated energy differences with more tight convergence criteria and a zoomed-in inset of the region around the minimum are shown here.

Figure R3. Plot of energy differences between total energy divided by number of units and the reference energy of 12-unit macrocycle with more tight convergence criteria. Inset shows the energy difference of neighboring points ($n=10,11,13,14$) relative to the global minimum ($n=12$).

However, as mentioned in point #2, the optimal stability for $n=12$ could be a result of multiple reasons such as geometry, thermodynamics and even aromaticity. Overall, based on our calculations, the 12-unit macrocycle exhibits the best stability from the thermodynamics point of view.

Actions: We have replaced the calculated energy differences with more tight convergence criteria as Fig. 1c in the main text and added Fig. R3 to SI as Fig. S3.

4. I wonder if the authors could discriminate the different bond orders in their BR-STM measurements of the 12-OQC? This is to check agreement with Clar's rule (disjoint aromatic pi sextets) on polycyclic aromatic hydrocarbons (PAH). AFM would have been a more useful technique here though.

We thank reviewer's suggestion on the implantation of AFM to determine the bond order. In general, AFM can be a more powerful technique than BR-STM in terms of structural determination. However, we think it is still very difficult to analyse the bond order of this monomer/macrocycle *via* AFM imaging. The "artificial distortion" in the AFM image of the periphery of a molecule is a well-known issue. The bond order analysis is typically performed for the interior part of a molecule based on previous studies (Science, 337, 6100 (2012)). It is thus expected that the bond order analysis will be less reliable since the distortion (as shown in the BR-STM image in Fig. 2a) cannot be avoided due to the relatively small size of our monomer. In addition, a strong distortion has been observed in the AFM image of a sister molecule: 3-triangulene (Nature Nanotech. 12, 308 (2017)).

5. On-surface debromination of organic adsorbates is known to occur readily on coinage metal surfaces even at room T. According to the experimental details, precursor 1 was

deposited at 360 K onto clean Au(111) surface held at room temperature.

I think on-surface debromination may occur here before or about the same time as the mechanism shown in Scheme 1.

Did authors track the debromination after precursor deposition? On STM, Br adatoms should appear as small protrusions on the surface near the debrominated precursor. A high-resolution STM image (if available) of the isolated (debrominated) precursor 1 on Au(111) could be helpful here to firmly establish the time ordering of the reaction events.

Figure R4. STM images of self-assembled Precursor 1, debrominated Precursor 1 and polymer chains after Ullmann coupling. **a**, STM image of self-assembled Precursor 1 on Au(111) without annealing. Br substituents appear as small dark dots as highlighted with white arrows ($V = 0.1$ V, $I = 300$ pA). **b**, STM image obtained after annealing Precursor 1 on Au(111) at 373 K reveals clusters of debrominated Precursor 1 (their nonplanar topology indicated the existence of hydrogens or methyl groups) coexisting with adsorbed Br atoms (highlighted with white arrows) near the edges of clusters ($V = -2.0$ V, $I = 300$ pA). **c**, STM image obtained after annealing Precursor 1 on Au(111) at 453 K reveals curved segments and polymer chains derived from incomplete cyclization as indicated by the small bright protrusions ($V = 1.0$ V, $I = 30$ pA).

We agree with reviewer that the debromination may occur before or about the same time as the formation of pentagonal rings. We did track the debromination process by annealing at different temperatures. We didn't observe small protrusions near the self-assembly of precursors (Fig. R4a) without post-annealing. STM image of self-assembled precursors reveals a peculiar feature consisting of two bright dots for individual molecular precursors. One smaller and darker dot could be contributed from the Br substituent (highlighted with white arrows in Fig. R4a) connected to the brighter dots. After post-annealing at 373 K for 15 minutes, we can find some clusters of debrominated precursors coexisting with Br adatoms near the edges of clusters as highlighted with white arrows (Fig. R4b). The observation of nonplanar topology of the debrominated precursors indicates that the cyclization process is likely to occur after the debromination. After post-annealing at 453 K for 15 minutes, most precursors have coupled into polymer chains or curved oligomers (Fig. R4c) *via* Ullmann-coupling. It is noted that the dehydrogenative cyclization is still not fully completed for some monomers, as evidenced by the small bright protrusions (hydrogens or tilted methyl groups). Annealing at 530 K for 20 minutes triggers the Ullmann coupling and dehydrogenative cyclization towards the formation of planar polymer chains, curved ring segments and perfect 12-unit circular polymer rings (Fig. 1b in the main text). Based on these observations, we can conclude that the debromination (onset temperature at around 373 K) occurs before the cyclization.

Actions: We have placed Fig. R4 to SI as Fig. S1 with the corresponding description.

Minor points:

=====

6. The transformation from precursor 1 to precursor 1' (Fig.1a) is some kind of Ullmann coupling reaction. Authors should mention this in the paper.

Yes, the transformation from precursor **1** to precursor **1'** includes Ullmann coupling. We have mentioned this in the manuscript.

Actions: We have highlighted the role of Ullmann coupling in polymerization in the revised manuscript, page 2, line 115.

7. Can authors comment on the adsorption geometry relative to the herringbone reconstruction & the planarity of 12-OQC? I assume that the 12-OQC macrocycle is physisorbed on Au(111). If so, what is the mean vertical height above top layer according to their DFT calculations? From Fig.S2a it seems that 12-OQC lies flat about 3 Å above top layer. The authors should probably give the DFT-relaxed structures of 12-OQC as a Supplementary Materials for the sake of reproducibility.

Figure R5. Side view (top) and top view (bottom) of the DFT-relaxed structure of 12-OQC on Au(111).

Based on our observation, the majority of 12-OQCs prefer to adsorb at the elbow sites. Our BR-STM image and DFT-relaxed structure suggest a planar structure of 12-OQC. Yes. The 12-OQC is physisorbed on Au(111) and the vertical height is about 3 Å above the top layer of Au(111). The side-view and top-view of DFT-relaxed structures of 12-OQC are shown in Fig. R5 above.

Actions: In view of reviewer's comments, we have included the DFT-relaxed structure of 12-OQC as Fig S4 in the SI.

8. I wonder whether a simple model of a quantum particle in an annular disk could be helpful

to understand the shape of the electronics states in Figs.2 f & g of the freestanding 12-OQC? The solution of the 2D-Schroedinger equation should be analytic yielding Bessel functions of the 1st kind.

In view of the comments from both reviewer 1 and 3, we also tested the analytical functions. The results show that these electronic states cannot be approximated by the model of a particle in an annular disk due to the following reasons.

(i) Solving the 2D Schrödinger's equation does indeed yield Bessel functions of the first and second kind. We used the asymptotic forms of the Bessel functions as well as the inner and outer radii of the 12-OQC to obtain $E_n \sim Ryd \times \left(\frac{n}{3}\right)^2$ as the approximate form of the energy spacing. From this, we observe that the energy separations between calculated levels are much bigger than the separations between, for example, the VB and CB of 12-OQC.

(ii) In addition, the experimental molecular states show a structure-dependent localization: for example, dI/dV maps (Fig. 2f,g in the main text) reveal a stronger intensity localized at the ZS (zigzag side) and PS (pentagon-decorated side) for CB and VB, respectively. The model of a smooth-walled annular disk fails to capture these features, as seen by the model's lower-lying eigenstates (see below).

Figure R6. Eigenstates of particle in an annular disk for $l = 0, 1$ and $n = 1, 2, 3$.

From our analysis, we feel that the model of a particle in an annular disk is not able to adequately describe the observed electronic states, which is most likely contributed from a collection of nearly degenerate molecular frontier orbitals.

9. In Fig.2 f & g, authors should give the calculated energy range of these DFT orbitals. Fig.S3 caption should give the energies of individual Kohn-Sham orbitals and provide the isosurface value of the representation.

We have included these information including the energy values of the calculated orbitals and the isosurface value to the revised manuscript.

ψ_1 : -0.68601 eV; ψ_2 : -0.67676 eV; ψ_3 : -0.65554 eV; ψ_4 : 1.03913 eV; ψ_5 : 1.05283 eV; ψ_6 : 1.06462 eV.

The frontier orbitals displayed in Fig. S7 (previous Fig. S3) were plotted with an isosurface value of $0.015 \text{ e}/\text{\AA}^3$.

Actions: We have modified Fig. 2 and Fig. S7 (previous Fig. S3) and their captions to include the energy range and isosurface value.

10. Did the authors cite all relevant literature about the use of organic molecules in quantum corrals?

On a quick Google search I found (for example) J. Phys. Chem. Lett 7 (16), 3073 (2016) which was not cited here.

Actions: We have cited these references (Nat. Nanotechnol. 2, 99 (2007); J. Phys. Chem. Lett 7 (16), 3073 (2016).) as Ref [12] and [44] in the revised manuscript.

11. Perhaps an electron localization function (ELF) plot of the freestanding 12-OQC would be helpful for point #2.

Thanks for this suggestion. However, as discussed in the response to Point #2, monomer and macrocycle have similar NICS values and localized aromaticity which means that the macrocycle does not show global aromaticity due to weak conjugation. Most likely, aromaticity plays a rather limited role here to stabilize the 12-unit macrocycle. We assume that the ELF plot will not provide much helpful information.

12. AFM measurements/simulations would have made this study more complete as is usual in this field of surface science. Any comment from the authors about this?

As mentioned in our response to Point #4, we think it is very difficult to analyse the bond order of this monomer/macrocycle *via* AFM imaging, although it can be a powerful technique than BR-STM imaging in some cases. The “artificial distortion” in the AFM image of the periphery of a molecule is a common issue. The bond order analysis is typically performed for the interior part of a molecule based on previous studies (Science, 337, 6100 (2012)). It is thus expected that the bond order analysis will be less reliable since the distortion (as shown in the BR-STM image in Fig. 2a) cannot be avoided due to the relatively small size of our monomer. In addition, a strong distortion has been observed in the AFM image of a sister molecule: 3-triangulene (Nature Nanotech. 12, 308 (2017)). We think the chemical structure of 12-OQC has been probed clearly by BR-STM, which works equally as AFM to identify chemical structure.

REVIEWERS' COMMENTS

Reviewer #1 (Remarks to the Author):

I am satisfied with the response of the authors to my queries and I consider the manuscript fit for publication.

Reviewer #3 (Remarks to the Author):

The authors have done an excellent work here and answered well the comments.

The revised manuscript is now ready for publication in Nat. Comm.

Thank you.